# How Does Land Fragmentation Affect Agricultural Technical Efficiency? Based on Mediation Effects Analysis

Chunfang Zhou [1], Yuluan Zhao [1,*], Mingshun Long [1] and Xiubin Li [2]

1    School of Geography and Environmental Science, Guizhou Normal University, Guiyang 550025, China
2    Institute of Geographic Sciences and Natural Resources Research, Chinese Academy of Sciences, Beijing 100101, China
*    Correspondence: zhaoyl@gznu.edu.cn; Tel.: +86-151-8695-0727

**Abstract:** The scientific revelation of the mechanism underlying land fragmentation's influence on agricultural technical efficiency is extremely important. This study utilized survey data from 305 villages across 12 provinces in Southern China in 2020 to assess technical efficiency through the application of the stochastic frontier production function. Moreover, we investigated the direct impact of land fragmentation on technical efficiency and the indirect impact transmitted through crop diversification and part-time farming by employing Tobit and mediating effect models, respectively. The key findings are as follows: (1) The sampled farmers, on average, operated 0.614 hectares of land with 17.395 plots, and the mean of their technical efficiency was 0.630. (2) The overall effect of land fragmentation on technical efficiency demonstrated a "U"-shaped relationship. (3) Crop diversification and part-time farming were mediating factors in the impact of land fragmentation on technical efficiency. Specifically, an "inverted U"-shaped relationship existed between land fragmentation and crop diversification, whereas a negative linear relationship was observed between land fragmentation and part-time farming. Conversely, crop diversification presented a positive linear relationship with technical efficiency, and part-time farming had an "inverted U"-shaped relationship with technical efficiency. (4) The impact of land fragmentation on technical efficiency varied across altitude zones. It is recommended to control land fragmentation based on local conditions, encourage crop diversification, and strengthen employment guidance and skills training for farmers to ensure the orderly transfer of land.

**Keywords:** land fragmentation; technical efficiency; crop diversification; part-time farming; Southern China

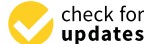



## 1. Introduction

As a major agricultural and populous country, China provides sustenance for 22% of the global population, with only 7% of the globally cultivated land, while simultaneously facing significant challenges in food production [1]. Although food production has maintained steady growth for more than a decade, there is still a structural imbalance [2], resulting in the need to import large quantities of soybeans, cereals, and potatoes [3]. Furthermore, the global food situation in recent years has become increasingly pessimistic due to the combined effects of extreme weather events, geopolitical shocks, and the COVID-19 pandemic [4,5]. Consequently, several countries have suspended or restricted their food exports [6]. In light of these circumstances, it is imperative for China to prioritize its food security by adhering to a strict cropland protection system [2], tapping the potential of the comprehensive utilization of saline land and other reserve land to expand the area of arable land [7]. Furthermore, implementing ecosystem restoration projects to mitigate and reverse land degradation [8], optimizing the allocation of agricultural production resources, and enhancing the agricultural technical efficiency are also vital [9]. The current objective of promoting agricultural land transfer and consolidation in China is to optimize

land resource allocation and reduce cultivated land fragmentation, with the objective of improving agricultural production efficiency [10,11].

Land fragmentation refers to the challenge of obtaining contiguous and concentrated cultivated land, which is evidenced by diverse small plots and their spatial dispersion [12,13]. This land use pattern is not consistent with large-scale management [14] and can potentially jeopardize agricultural operations [15]. In China, where there are many people with limited land, the household contract responsibility system has effectively motivated farmers to participate in agricultural production by dividing land equally among the population, according to the quality of the land and the distances of the plots to their homes [16]. However, this system has resulted in a fragmented land use pattern due to the strict household registration system and scarcity of non-agricultural employment opportunities [17,18]. Consequently, farmers often engage in self-exploitation to maintain their livelihoods and increase land output rates through intensive labor input, leading to agricultural intensification via involutional agricultural practices, such as crop rotation systems [19,20]. Land fragmentation has facilitated the cultivation of a variety of crops with different adaptive capacities, such as soil type, slope, and microclimatic variations, which, in turn, promotes the diversification of cultivated crops, disperses the demand for labor, and reduces production and price risks [21–23]. However, with the implementation of the socialist market mechanism in the early 1990s, driven by the high income of urban non-agricultural employments, the scale of peasant migrant work has gradually increased. Farmers in China commonly engage in part-time farming, which is beneficial for risk avoidance in agricultural production, optimization of labor force allocation efficiency, and maximization of family income [24]. Nevertheless, part-time farming may lead to a shortage of agricultural labor, resulting in a decline in agricultural productivity, which is exacerbated by the fragmentation of agricultural land [25–27].

The relationship between land fragmentation and agricultural technical efficiency has been extensively researched. However, the findings of these studies are not uniform. Numerous studies have demonstrated a strong negative correlation between cropland fragmentation and technical efficiency. The diverse plots operated by farmers, the scattered fragmentation of these plots, and the small area of individual plots lead to the inefficient allocation of agricultural production factors, impede the adoption of new agricultural technologies, and reduce the technical efficiency [28–32]. Furthermore, a 1% increase in land fragmentation is associated with a 0.05% decrease in rice output and 0.03% decrease in technical efficiency [33]. In contrast, several studies reported a significant positive correlation between land fragmentation and technical efficiency. These studies suggest that having a high number of plots operated by farmers promotes diversified cultivation, disperses farmers' agricultural production risks, and enhances agricultural production efficiency [34,35]. In addition, certain studies have indicated that the relationship between land fragmentation and technical efficiency is complex and not simply linear [36,37].

Agricultural production and management activities involve the aggregation and allocation of factors such as labor and capital on land. The fragmentation of cultivated land can have a significant impact on the decision-making and management practices of farm households regarding the allocation of these production factors [29]. First is the impact on the allocation of labor factors of farm households. The large number and spatial dispersion of plots increase the transportation time of farmers to various non-adjacent plots and from each plot to their homes, while the small size of plots is not conducive to mechanized operations. Consequently, the use of mechanized alternative labor is weakened. Farmers may need to increase the time spent on agricultural labor and reduce the supply of non-agricultural labor to maintain the same level of agricultural output. This can reduce the degree of part-time employment in farm households [25]. Moreover, the impact of part-time farming on agricultural production efficiency is a matter of concern [38,39]. Second, farmers' planting decisions are also affected. Land fragmentation encourages farmers to diversify their crops, and the fragmentation and distribution of land parcels enable farmers

to plant various crops based on the agricultural conditions and soil types of different land parcels. This diversification can affect the efficiency of agricultural production [40,41].

In summary, numerous studies have been conducted on the impact of land fragmentation on agricultural technical efficiency, yielding a wide range of findings. However, there are several limitations to this body of research. First, the results are characterized by considerable heterogeneity, with regional variability being one of the primary sources. Scholars typically focus on the average effect of land fragmentation on the overall agricultural production efficiency of a particular region rather than examining how it varies across different regions. This is because of differences in natural resource endowment, the level of development of the rural factor market, and the structure of agricultural planting, all of which play vital roles in the impact of land fragmentation on efficiency. Second, there has been insufficient research on the mechanisms of the impact. The proposed policy has not been targeted as its implementation has not been informed by previous research, which has only assessed the general effect of land fragmentation on the technical efficiency. The intermediate variables that connect both have not undergone adequate investigation. Hence, this study focuses on Southern China as the research area due to its complex and diverse geomorphology, as well as its high degree of land fragmentation. Furthermore, the area exhibits variations in the structure of agricultural cultivation and the level of economic development, making it representative and avoiding sample homogeneity. Moreover, this study utilizes the stochastic frontier production function to calculate the agricultural technical efficiency of farmers. Additionally, planting diversification and part-time farming are considered as mediating variables to explore the mechanisms of land fragmentation on agricultural technical efficiency. The findings from this research provide valuable insights for managing land fragmentation and enhancing agricultural technical efficiency.

## 2. Materials and Methods

### 2.1. Study Areas

The area of this study is Southern China, encompassing 12 provinces, including Zhejiang, Anhui, Hubei, Jiangxi, Hunan, Fujian, Guangdong, Guangxi, Guizhou, Chongqing, Sichuan, and Yunnan, with geographic coordinates ranging from $97°21'$ to $122°49'$ E and $20°13'$ to $34°39'$ N. The region is characterized by a subtropical monsoon climate, with predominant soil types, including red loam, yellow loam, and rice soil. Agricultural land, such as cropland, garden land, and forest land, dominates the land use in this area. According to the Third National Land Survey of China, the arable land, garden land, and forest land areas in the study area in 2019 were 40,063.85, 11,615.87, and 144,582.79 thousand hectares, respectively. Compared to 2009, the arable land area witnessed a decrease of 8646.36 thousand hectares, while the garden land and forest land areas saw an increase of 2977.29 and 13,799.60 hectares, respectively. The region is characterized by a diverse range of landforms, and altitude had a direct impact on the water–heat mix in the region, which, in turn, affected agricultural productivity. The region was divided into low-altitude zones (200 m above sea level), medium-altitude zones (from 200 to 1000 m), and high-altitude zones (>1000 m) based on the altitude. The data utilized in this study come from field research conducted by the group from July to August 2020 across the 12 provinces in Southern China. A total of 305 administrative villages were selected by using stratified and random sampling methods (Figure 1). Four to six farmers were randomly surveyed in each village, with 1661 questionnaires collected. Among them, 1498 were valid questionnaires (299 from low-elevation zones, 913 from middle-elevation zones, and 286 from high-elevation zones). The questionnaire indicators were comprehensive and representative, covering farm household characteristics, such as farmland resource status, cropland cultivation, and family size and income in 2019.

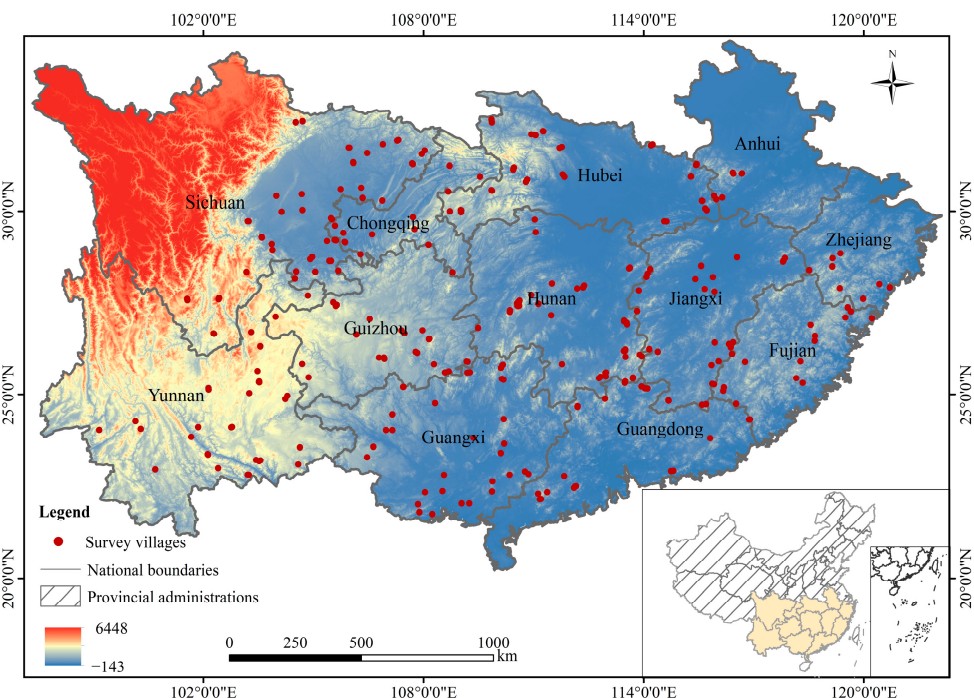

**Figure 1.** Geographic distribution of surveyed villages.

## *2.2. Model Specification*

### 2.2.1. Stochastic Frontier Production Function

To examine the impact of land fragmentation on agricultural technical efficiency, it is imperative to first quantify agricultural technical efficiency. The non-parametric approach of data envelopment analysis (DEA) and the parametric method of stochastic frontier production function (SFA) are the most commonly used techniques for evaluating agricultural technical efficiency [42]. However, due to the diverse natural conditions and social environments in the survey area, as well as variations in respondents' cognitive abilities and accuracy in answering questions, this study employed the SFA model to measure technical efficiency [43]. The model is as follows:

$$Y_i = f(X_i, \beta) TE_i e^{v_i} \tag{1}$$

where $Y_i$ is the agricultural output of farmer $i$; $X_i$ is the agricultural input of farmer $i$; $\beta$ is an unknown parameter; $TE_i$ is the level of agricultural technical efficiency of farmer $i$, which satisfies $0 < TE_i \leq 1$. If $TE_i = 1$, farmer $i$ is in the frontier of agricultural technical efficiency. $e^{v_i}$ is the stochastic shock to the agricultural production of farmer $i$. Assuming that $f(X_i, \beta) = e^{\beta_0} X_{1i}^{\beta_1} \cdots X_{ki}^{\beta_k}$ (Cobb Douglas production function, $k$ is the number of agricultural production inputs of the $X_{ki}$ species), the logarithm of both sides of Equation (1) can be obtained:

$$lnY_i = \beta_0 + \sum_{k=1}^{k} \beta_k lnX_{ki} + lnTE_i + v_i \tag{2}$$

Given that $0 < TE_i \leq 1$, it follows that $lnTE_i \leq 0$. Define $u_i = -lnTE_i \geq 0$. Subsequently, Equation (2) can be written as:

$$lnY_i = \beta_0 + \sum_{k=1}^{k} \beta_k lnX_{ki} + v_i - u_i \tag{3}$$

where $v_i$ is the random error term, which is used to judge the influence of measurement error and random interference factors. It is assumed that $v_i$ follows independently and distributed identically, that is, $v_i \sim N(0, \sigma_v^2)$; $u_i$ is the inefficiency term, which is used to capture the technical inefficiency of random variables. Moreover, it is assumed that $u_i$ is uncorrelated with $v_i$ and follows a half-normal distribution, that is, $u_i \sim N^+(0, \sigma_u^2)$. The production efficiency of farmers $TE_i$ can be expressed as:

$$TE_i = Y_i / \exp(X_i, \beta) = \exp(X_i, \beta - u_i) / \exp(X_i, \beta) = \exp(-u_i) \tag{4}$$

### 2.2.2. Mediating Effect Model

Mediated effects modeling enables in-depth research of the process and mechanism of influence between factors, not only to explain the direct link between variables but also to investigate the indirect channel of action between variables [44,45]. Three models are constructed:

$$TE_i = a_0 + a_1 Fra_i + a_2 Z_i + \varepsilon_1 \tag{5}$$

$$Med_i = b_0 + b_1 Fra_i + b_2 Z_i + \varepsilon_2 \tag{6}$$

$$TE_i = c_0 + c_1 Fra_i + c_2 Med_i + c_3 Z_i + \varepsilon_3 \tag{7}$$

where $Fra_i$ represents land fragmentation; $Med_i$ represents the mediating variables (including crop diversification and part-time farming); $Z_i$ represents other control variables; $b_1 \times c_2$ represents the mediating effect size of fragmentation, which refers to the indirect impact of land fragmentation on agricultural technical efficiency, wherein the indirect effect occurs via its influence on crop planting structure and farmers' engagement in part-time occupations.

### 2.3. Descriptions of the Variables

### 2.3.1. Explained Variable

The explanatory variable in this study was agricultural technical efficiency (TE), which was calculated using Equation (4) of the previously discussed SFA model. The SFA model defined output Y as farmers' total agricultural production value, whereas input components X comprised the area of land (A), the quantity of capital inputs (C), such as seeds, fertilizers, pesticides, and equipment inputs, and the labor input (L) assessed on working days.

### 2.3.2. Core independent Variables

This study focused on land fragmentation (Fra) as the core independent variable, with the count of plots serving as the metric for measuring fragmentation according to the relevant concept.

### 2.3.3. Mediating Variables

Two mediating factors were considered: crop diversification (DP) and part-time farming (NF). The assessment of f crop diversification (DP) was based on the number of crops sown. The degree of part-time farming (NF) in rural households was gauged by the percentage of non-agricultural income, represented as the ratio of non-agricultural income to the total household income.

### 2.3.4. Control Variables

Control variables involved the attributes of household heads and traits. The attributes of the household head consisted of age (HA), health status (HH), and education level (HE). Furthermore, five indicators were selected to represent household traits, including the number of people in the household (FP), the number of laborers in the household (FL), the presence of disabled individuals (Di), the presence of Communist Party members (Co), and the presence of village leadership (VL). Table 1 displays the definitions, assignments, and overall descriptive statistics for the variables mentioned above.

**Table 1.** Variable setting and descriptive statistics.

| Variable | Variable Description | Mean | S. D |
|---|---|---|---|
| Y | Total agricultural income of farmers (yuan) | 26,306.553 | 169,925.454 |
| A | Area of land input in agricultural production (ha) | 0.614 | 2.589 |
| C | Capital input in agricultural production (yuan) | 10,812.559 | 102,480.138 |
| L | Labor input in agricultural production (days) | 328.779 | 1835.796 |
| TE | Agricultural technical efficiency, calculated by SFA model | 0.630 | 0.108 |
| Fra | Land fragmentation, measured by number of plots | 17.395 | 33.147 |
| DP | Number of crop types | 2.345 | 1.304 |
| NF | Proportion of non-farm income to household income | 0.683 | 0.323 |
| HA | Age of household head (years) | 56.760 | 10.767 |
| HH | Disabled = 0, very poor = 1, poor = 2, moderate = 3, good = 4 | 3.511 | 0.834 |
| HE | Illiteracy = 1, Primary school = 2, Middle school = 3, High school or vocational secondary school = 4, College and above = 5 | 2.485 | 0.908 |
| FP | Number of household members | 4.200 | 1.927 |
| FL | Number of household laborers | 2.964 | 1.273 |
| Di | Presence of disabled individuals in the household, No = 0, Yes = 1 | 0.193 | 0.395 |
| Co | Presence of Communist Party members in the household, No = 0, Yes = 1 | 0.149 | 0.356 |
| VL | Presence of village leadership in the household, No = 0, Yes = 1 | 0.009 | 0.093 |

## 3. Results

### 3.1. Land Fragmentation and Agricultural Technical Efficiency

3.1.1. Land Fragmentation Degree

The average operational area of farm families in the research region was 0.614 hectares, and the mean number of cultivated plots was 17.395, with 61.95% of farm households having fewer than 12 plots (Figure 2a). There was no statistically significant difference between different altitude zones in terms of area of operation and the degree of land fragmentation, with the mean values for area of operation of farm households in low-, medium-, and high-altitude zones being 0.741 ha, 0.511 ha, and 0.812 ha, and the medians being 0.253, 0.260, and 0.430 ha, respectively. And the average numbers of plots of farm households in low-, medium-, and high-altitude zones were 15.351, 18.499, and 16.007 (Figure 2b), respectively, with median values of 8, 10, and 10.

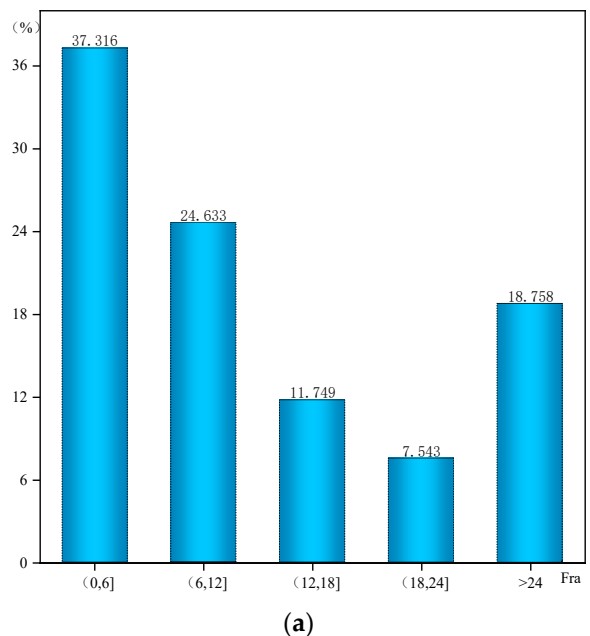

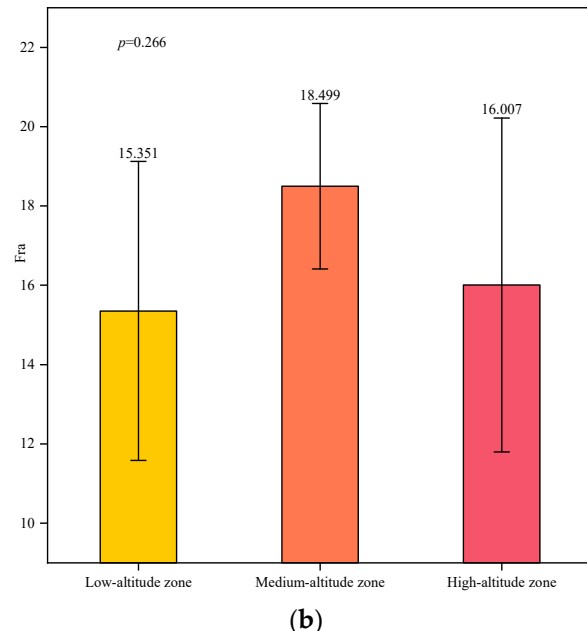

**Figure 2.** (**a**) Land fragmentation among farmers; (**b**) fragmentation in different altitude regions.

3.1.2. Assessment of Agricultural Technical Efficiency

The agricultural technical efficiency in Southern China was assessed using Stata17.0 software. The result, $\gamma = \sigma_u{}^2 / (\sigma_u{}^2 + \sigma_v{}^2) = 0.522$, indicated that the dominant source of technical inefficiency was the stochastic perturbation term (Table 2). Consequently, it was deemed appropriate to employ a stochastic frontier production function.

**Table 2.** Test results of stochastic frontier production function model.

| Variable | Coefficient | Standard Errors | Z | $p > |z|$ |
|---|---|---|---|---|
| constant | 4.315 | 0.123 | 34.950 | 0.000 |
| lnA | 0.371 | 0.027 | 13.500 | 0.000 |
| lnC | 0.526 | 0.020 | 25.970 | 0.000 |
| lnL | 0.098 | 0.014 | 6.900 | 0.000 |
| $\sigma_v$ | 0.587 | 0.023 | | |
| $\sigma_u$ | 0.614 | 0.059 | | |
| $\sigma^2$ | 0.722 | 0.055 | | |
| Log likelihood | 18.26 | | | |

The study area revealed that the minimum, maximum, and mean values of the technical efficiency of farmers' production were 0.036, 0.855, and 0.630, respectively. Additionally, 59.95% of the farmers exhibited technical efficiency within a range of 0.6 to 0.8 (Figure 3a). Statistically significant disparities existed in the technical efficiency across various altitudinal zones. Specifically, producers operating in low-, medium-, and high-altitude zones achieved agricultural technical efficiencies of 0.621, 0.638, and 0.612, respectively (Figure 3b).

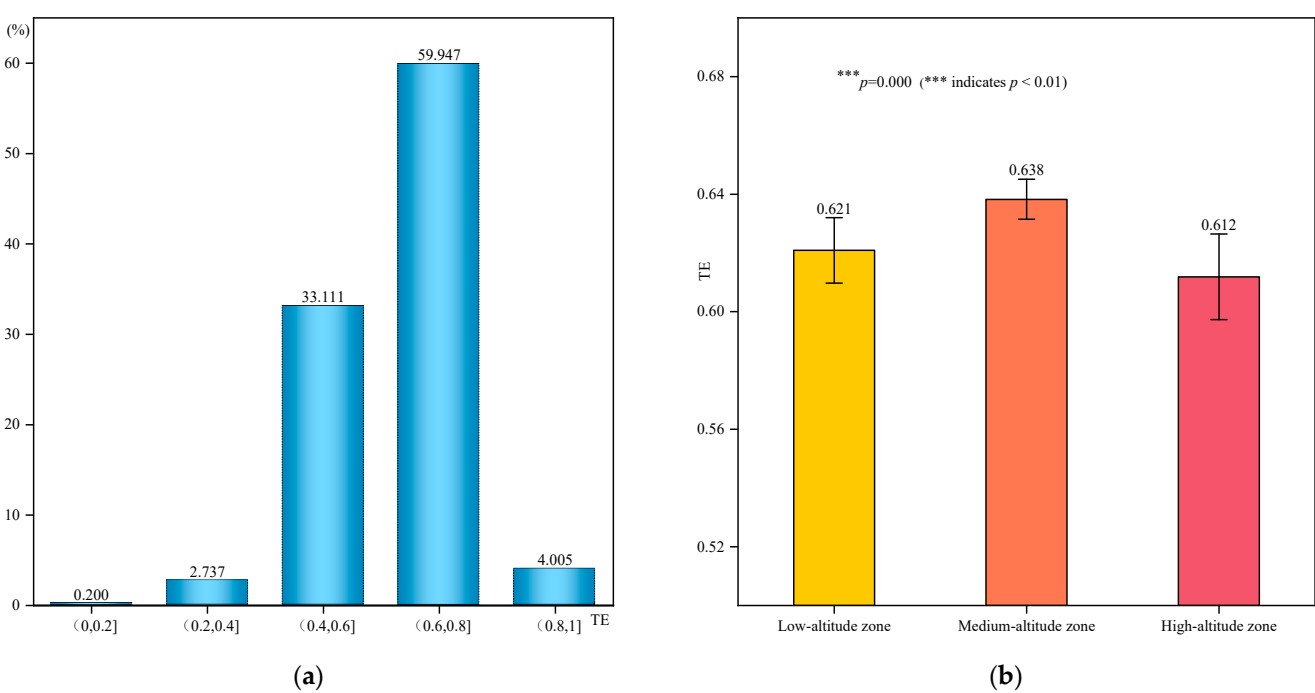

(a)   (b)

**Figure 3.** (**a**) Technical efficiency of farmers; (**b**) technical efficiency in different altitude regions.

A cross-study analysis was conducted to determine the preliminary association between the variables of land fragmentation, agricultural technical efficiency, diversity of farmers' planting, and part-time farming. Figure 4 shows that the agricultural technical efficiency of farmers initially decreased and then increased as the degree of land fragmentation increased. Planting diversification exhibited an inverted U-shaped relationship with the degree of fragmentation. It initially increased and subsequently decreased as the degree

of fragmentation increased. Furthermore, the degree of part-time farming decreased as the degree of land fragmentation increased.

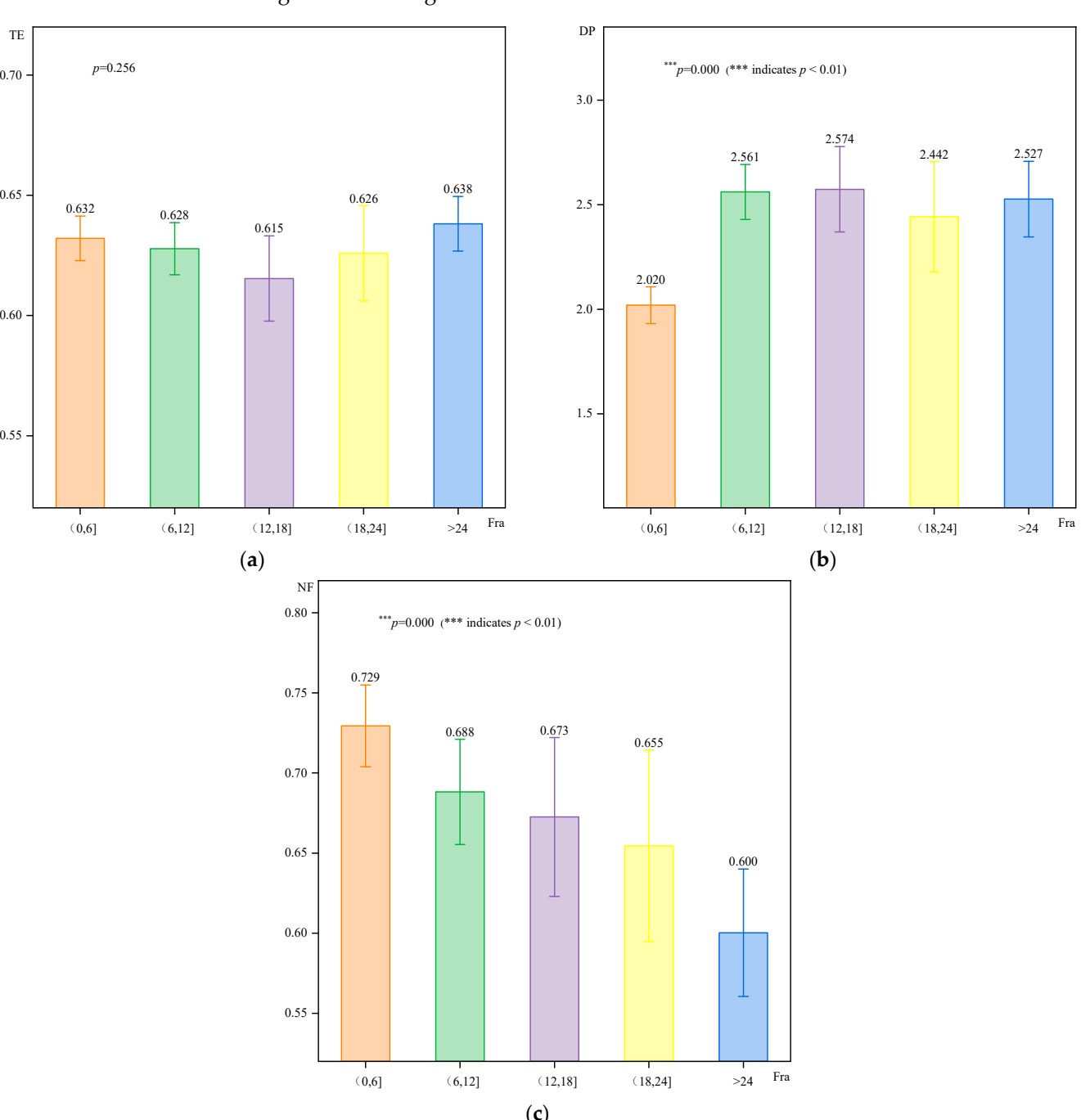

**Figure 4.** (**a**) Cross analysis of fragmentation and agricultural technical efficiency; (**b**) cross analysis of fragmentation and crop diversification; (**c**) cross analysis of fragmentation and part-time farming.

### 3.2. Overall Effect of Land Fragmentation on Agricultural Technical Efficiency

The Tobit regression model was used in Equation (5) to elucidate the overall impact of land fragmentation on agricultural technical efficiency. Data were consolidated to ensure the scientific validity of the mediating effect model. According to Models 1 and 2 (Table 3), the estimated coefficient of the first power of land fragmentation was 0.018, which was statistically significant at the 5% level. The calculated coefficient of quadratic fragmentation was 0.003, which was statistically significant at the 10% level, indicating that the impact of land fragmentation on agricultural technical efficiency had a weak "U" curve relationship.

**Table 3.** Overall effect of land fragmentation on agricultural technical efficiency.

| Variable | Total | | Low-Altitude Zone | | Medium-Altitude Zone | | High-Altitude Zone | |
|---|---|---|---|---|---|---|---|---|
| | Model 1 | Model 2 | Model 3 | Model 4 | Model 5 | Model 6 | Model 7 | Model 8 |
| lnFra | −0.002 (−0.882) | −0.018 ** (−2.138) | −0.009 * (−1.713) | −0.012 (−0.743) | −0.002 (−0.755) | −0.024 ** (−2.381) | 0.007 (0.869) | 0.008 (0.312) |
| (lnFra)$^2$ | | 0.003 * (1.959) | | 0.001 (0.198) | | 0.005 ** (2.258) | | −0.0001 (0.027) |
| lnHA | −0.018 (−1.208) | −0.018 (−1.173) | 0.020 (0.625) | 0.021 (0.644) | −0.041 ** (−2.055) | −0.041 ** (−2.063) | −0.023 (−0.642) | −0.023 (0.643) |
| HH | 0.012 *** (3.478) | 0.011 *** (3.347) | 0.014 ** (2.198) | 0.014 ** (2.151) | 0.014 *** (3.167) | 0.013 *** (2.998) | 0.001 (0.159) | 0.001 (0.157) |
| HE | 0.004 (1.251) | 0.004 (1.186) | −0.008 (−1.180) | −0.008 (−1.182) | 0.003 (0.797) | 0.003 (0.767) | 0.010 (1.114) | 0.010 (1.115) |
| lnFP | −0.007 (−0.833) | −0.007 (−0.851) | −0.020 (−1.181) | −0.020 (−1.190) | −0.008 (−0.748) | −0.009 (−0.789) | 0.005 (0.208) | 0.005 (0.206) |
| lnFL | 0.018 * (1.796) | 0.018 * (1.849) | 0.020 (0.986) | 0.020 (1.002) | 0.021 * (1.735) | 0.021 * (1.747) | 0.009 (0.334) | 0.009 (0.334) |
| Di | −0.016 * (−1.844) | −0.016 * (−1.812) | −0.021 (−1.227) | −0.021 (−1.216) | −0.018 * (−1.725) | −0.018 * (−1.763) | 0.004 (0.171) | 0.004 (0.168) |
| Co | 0.042 *** (4.323) | 0.041 *** (4.285) | 0.048 ** (2.186) | 0.048 ** (2.189) | 0.044 *** (3.879) | 0.043 *** (3.871) | 0.014 (0.490) | 0.014 (0.490) |
| VL | −0.012 (−0.408) | −0.012 (−0.394) | 0.003 (0.055) | 0.003 (0.070) | −0.024 (−0.607) | −0.027 (−0.681) | −0.041 (−0.324) | −0.041 (−0.324) |
| log likelihood | 1231.63 *** | 1233.55 *** | 279.72 ** | 279.74 * | 784.63 *** | 787.18 *** | 192.07 | 192.07 |
| N | 1498 | 1498 | 299 | 299 | 913 | 913 | 286 | 286 |

Note: t-values in parentheses, * indicates $p < 0.10$, ** indicates $p < 0.05$, and *** indicates $p < 0.01$.

The effect of the household head's health status on agricultural technical efficiency, while controlling for other variables, was 0.011 and statistically significant at the 1% level. Concerning household characteristics, the family labor force exhibited a regression coefficient of 0.018, which was significant at the 10% level. Additionally, the presence of disabled family members was associated with a regression coefficient of −0.016, which was significant at the 10% level. Finally, the presence of party members in the family was associated with a regression coefficient of 0.041, which was significant at the 1% level.

Based on the results presented for Models 3 through 8 (Table 3), the impact of fragmented cultivated land on agricultural technical efficiency varied significantly across regions with different resource endowments and altitudes. Regression analysis revealed that cropland fragmentation had a statistically significant negative linear impact on agricultural technical efficiency in low-altitude regions (coefficient = −0.009, $p = 0.088$). In the middle-altitude region, the relationship between agricultural technical efficiency and cropland fragmentation displayed a U-shaped curve, with the first power coefficient being −0.024 and the second power coefficient being 0.005, both of which were significant at the 5% level. However, the regression coefficient for agricultural technical efficiency in the high-altitude region was positive without meeting the criteria for statistical significance. This suggested that land fragmentation did not substantially affect agricultural technical efficiency in the high-altitude region. Therefore, a separate study on the mediating effect was not necessary for high-altitude regions.

### 3.3. Impact of Land Fragmentation on Mediating Variables

Based on the regression results (Table 4) of the analysis examining the relationship between land fragmentation and the mediating variables of crop diversification and part-time farming, it can be observed that, with regard to crop diversification, the estimated

coefficient for the primary aspect of land fragmentation was 0.281, and the estimated coefficient for the quadratic aspect was $-0.040$, both of which passed the significance test at the 1% statistical level, suggesting that land fragmentation displayed a significant "inverted U" curve relationship with crop diversification. The fragmentation of farmland may encourage farmers to cultivate a variety of crops based on the distinct water and heat conditions, as well as the soil quality of individual plots, thereby promoting planting diversification. However, when the critical value was exceeded, the complexity of managing and cultivating the farm increased, which, in turn, negatively affected cultivation diversification. For part-time farming, there was a statistically significant negative linear relationship between land fragmentation and part-time farming, with a regression coefficient of $-0.065$, which was significant at the 1% level. This means that farmland fragmentation had a significant inhibitory effect on part-time farming, as it increased agricultural labor time and reduced the time available for non-farm activities.

**Table 4.** Effects of land fragmentation on mediating variables.

| Variable | Total | | Low-Altitude Zone | | Medium-Altitude Zone | |
|---|---|---|---|---|---|---|
| | **lnDP** | **NF** | **lnDP** | **NF** | **lnDP** | **NF** |
| lnFra | 0.281 *** | −0.065 *** | 0.272 *** | −0.140 *** | 0.283 *** | −0.044 *** |
| | (6.891) | (−8.823) | (3.100) | (−9.281) | (5.339) | (−5.068) |
| $(lnFra)^2$ | −0.040 *** | | −0.033 * | | −0.039 *** | |
| | (−4.794) | | (−1.87) | | (−3.574) | |
| control | Yes | Yes | Yes | Yes | Yes | Yes |
| log likelihood | −1163.58 *** | −329.66 *** | −216.19 *** | −26.95 *** | −736.96 *** | −147.67 *** |
| N | 1498 | 1498 | 299 | 299 | 913 | 913 |

Note: t-values in parentheses, * indicates $p < 0.10$, ** indicates $p < 0.05$, and *** indicates $p < 0.01$.

The effects of land fragmentation on the two mediating variables of crop diversification and farm households exhibited consistent patterns across varying altitudes. However, the intensity of these effects varied, particularly with respect to farming households. Specifically, the regression coefficients for land fragmentation's impact on part-time farm households were $-0.140$ at low altitudes and $-0.044$ at medium altitudes. This suggested that the impact of land fragmentation on part-time farm households was significantly stronger in low-altitude areas than in medium-altitude areas.

### 3.4. Mediating Effect Analysis of Land Fragmentation on Agricultural Technical Efficiency

3.4.1. Mediating Effect of Crop Diversification

From Models 1 to 2 (Table 5), a positive linear relationship was observed between crop diversification, the mediating variable, and agricultural technical efficiency. The regression coefficient was 0.020, which was significant at the 1% level. By cultivating various crops according to the soil, location, and other conditions of each plot, farmers can enhance agricultural technical efficiency. The addition of crop diversification as a mediating variable did not eliminate the influence of land fragmentation on agricultural technical efficiency, as the influence coefficient still passed the significance test. This suggested that crop diversification had a significant mediating effect on the relationship between land fragmentation and agricultural technical efficiency, whereas it was not a complete mediating effect but rather a partial one.

**Table 5.** Mediating effect of crop diversification.

| Variable | Total | | Low-Altitude Zone | | Medium-Altitude Zone | |
|---|---|---|---|---|---|---|
| | Model 1 | Model 2 | Model 3 | Model 4 | Model 5 | Model 6 |
| lnFra | −0.023 *** (−2.781) | −0.022 *** (−2.681) | −0.012 ** (−2.092) | −0.012 ** (−2.088) | −0.028 *** (−2.728) | −0.026 ** (−2.602) |
| (lnFra)$^2$ | 0.004 ** (2.420) | 0.004 ** (2.301) | | | 0.005 ** (2.503) | 0.005 ** (2.355) |
| lnDP | 0.020 *** (3.799) | −0.007 (−0.506) | 0.020 * (1.811) | 0.055 * (1.761) | 0.013 ** (2.170) | −0.016 (−0.957) |
| (lnDP)$^2$ | | 0.019 ** (2.009) | | −0.027 (−1.208) | | 0.020 * (1.881) |
| control | Yes | Yes | Yes | Yes | Yes | Yes |
| log likelihood | 1240.73 *** | 1242.75 *** | 281.35 ** | 282.08 ** | 789.52 *** | 791.29 *** |
| N | 1498 | 1498 | 299 | 299 | 913 | 913 |

Note: t-values in parentheses, * indicates $p < 0.10$, ** indicates $p < 0.05$, and *** indicates $p < 0.01$.

Models 3 to 6 (Table 5) indicate that crop diversification had a consistently positive impact on agricultural technical efficiency across different altitude zones. Land fragmentation exhibited a statistically significant negative impact on agricultural technical efficiency in low-altitude areas, with the impact coefficient changing from −0.009 to −0.012 at the 5% significance level. This suggested that land fragmentation had a linear negative direct impact on agricultural technical efficiency and that crop diversification served as a mediating factor in the relationship between both. In the intermediate altitude region, the first-order regression coefficient for cultivated land fragmentation demonstrated a slight change from −0.023 to −0.028, while the quadratic coefficient also experienced a minor change from 0.004 to 0.005. Both coefficients passed the significance test, suggesting that crop diversification in this area partially mediated the relationship between land fragmentation and agricultural technical efficiency.

### 3.4.2. Mediating Effect of Household Part-Time Farming

Models 1 to 2 (Table 6) indicate that the mediating variable of part-time farming on agricultural technical efficiency exhibited an inverted "U" curve. The primary coefficient of part-time farming was 0.119, and the quadratic coefficient was −0.200, both of which were significant at the 1% level, indicating that agricultural technical efficiency initially increased and then decreased with an increase in part-time farming. When the degree of part-time farming was very low, farm household income mainly relied on agricultural income, and family labor primarily worked as casual workers in the vicinity during farm leisure time. Extra income can increase capital and technical inputs in agricultural production, thereby improving production efficiency. Conversely, when part-time farming exceeded the critical value, as it increased, farm households' usage of labor for agricultural production dropped, and the quality of labor utilized for agricultural production declined as young and able-bodied laborers migrated to work, resulting in a decline in agricultural productivity. The regression coefficient of farmland fragmentation remained significant after controlling for the mediator variable of part-time farming, suggesting that part-time farming partially mediated the impact of land fragmentation on agricultural technical efficiency.

From Models 3 to 6 (Table 6), the path for the impact of part-time farming on agricultural technical efficiency was consistent across different altitudes, exhibiting an inverted "U" curve. Additionally, the mediating effect of part-time farming on the relationship between land fragmentation and agricultural technical efficiency followed a consistent direction.

**Table 6.** Mediating effect of part-time farming.

| Variable | Total | | Low-Altitude Zone | | Medium-Altitude Zone | |
|---|---|---|---|---|---|---|
| | Model 1 | Model 2 | Model 3 | Model 4 | Model 5 | Model 6 |
| lnFra | −0.020 ** (−2.535) | −0.024 *** (−3.020) | −0.021 ** (−3.596) | −0.023 *** (−3.923) | −0.022 ** (−2.284) | −0.025 ** (−2.642) |
| $(lnFra)^2$ | 0.003 (1.611) | 0.003 * (1.936) | | | 0.003 (1.695) | 0.004 * (1.905) |
| NF | −0.088 *** (−9.922) | 0.119 *** (3.365) | −0.087 *** (−4.314) | 0.145 * (1.904) | −0.094 *** (−8.153) | 0.131 ** (3.087) |
| $NF^2$ | | −0.200 *** (−6.032) | | −0.220 *** (−3.157) | | −0.214 *** (−5.504) |
| control | Yes | Yes | Yes | Yes | Yes | Yes |
| log likelihood | 1281.22 *** | 1299.20 *** | 288.75 *** | 293.65 *** | 819.26 *** | 834.16 *** |
| N | 1498 | 1498 | 299 | 299 | 913 | 913 |

Note: t-values in parentheses, * indicates $p < 0.10$, ** indicates $p < 0.05$, and *** indicates $p < 0.01$.

*3.5. Robustness Tests*

3.5.1. Regression Model Stability Test

To evaluate the stability of the estimated results from the Tobit model, a linear regression model was utilized as a control variable, and the impact of land fragmentation on agricultural technical efficiency was re-estimated. The results demonstrated that the estimation results for each variable were largely consistent with those obtained from the Tobit model, with high altitudes serving as the reference group (Table 7). The coefficients for the low-altitude area, excluding Model 1, did not pass the significance test, whereas others did. All the coefficients for the middle-altitude area passed the 1% significance level test. These results suggested that variations in geomorphological types can lead to differences in the influence of cultivated land fragmentation on agricultural technical efficiency. In summary, the Tobit regression model demonstrated robust estimation results.

3.5.2. U-Shaped Test

Based on the regression model construction previously described, it has been indicated that land fragmentation, agricultural technical efficiency, crop diversification, and part-time farming all exhibit either a "U"-shaped or inverted "U"-shaped relationship. To enhance the reliability of the findings, this study utilized Stata to analyze the "U"-shaped correlation between variables. The findings indicated that in the assessment of the overall impact of fragmentation on agricultural technical efficiency, the model demonstrated an initial negative slope followed by a subsequent positive slope (Table 8). However, the positive slope fails to meet the 10% significance level, suggesting the presence of a weak "U"-shaped relationship between the two variables. Additionally, the test of the inverse "U"-shaped relationship between fragmentation and crop diversification revealed that the established model tended to be initially positive and then negative, with both values being significant at the 1% level, indicating a significant inverse "U"-shaped relationship between the two variables. Furthermore, in the examination of the inverse "U"-shaped relationship between simultaneous farming by farmers and agricultural technical efficiency, the set oblique was initially positive and then negative, with both values being significant at the 1% level, suggesting a significant inverse "U"-shaped relationship between the two variables. Combined with the "U" test results (Figure 5), the aforementioned conclusion regarding the existence of a "U"-shaped relationship between variables was robust.

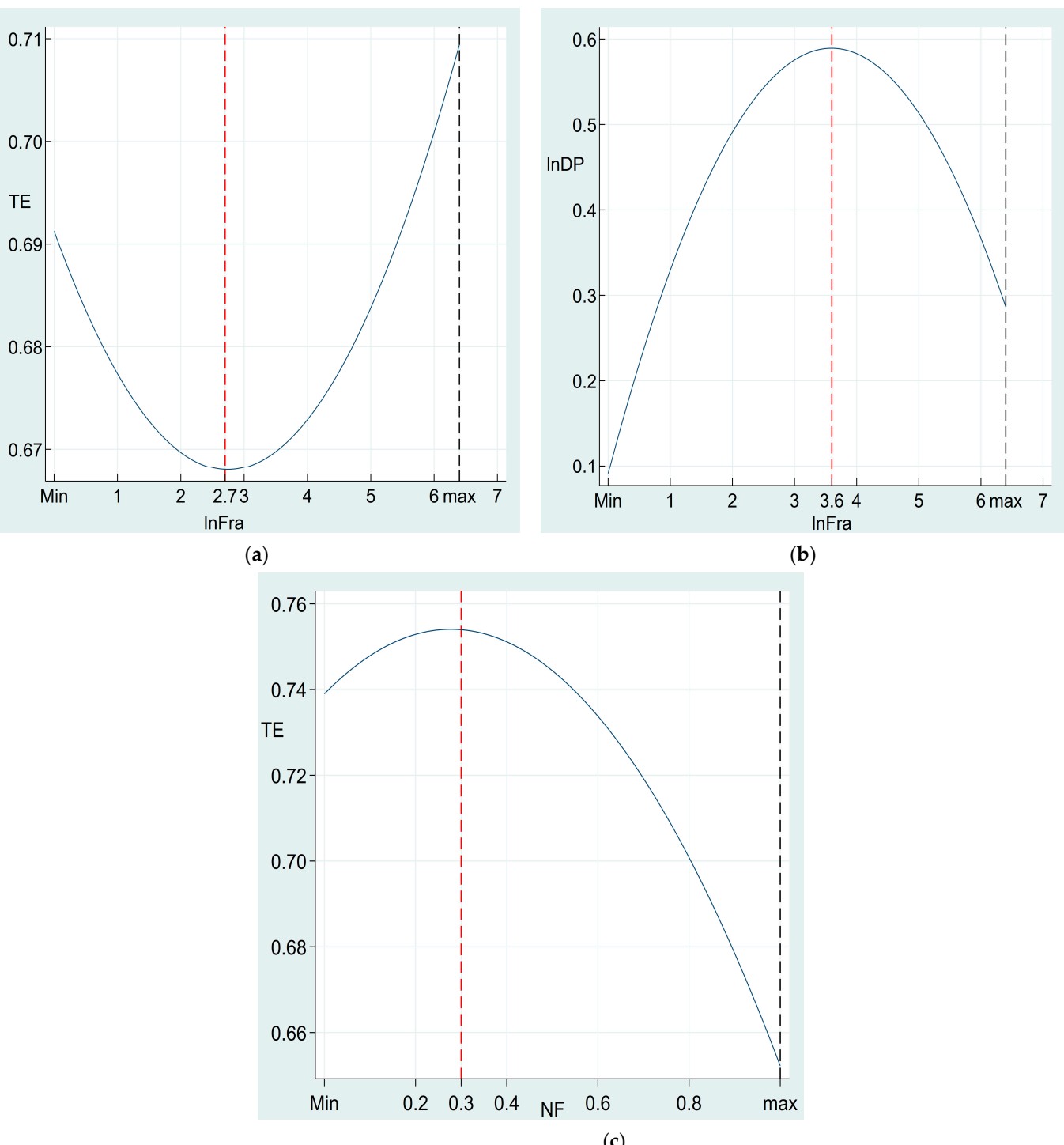

**Figure 5.** (**a**) U-relationship test for land fragmentation and technical efficiency; (**b**) inverted U-relationship test for land fragmentation and crop diversification; (**c**) inverted U-relationship test for part-time farming on technical efficiency.

**Table 7.** Results of the stability test.

| Variable | TE | lnDP | NF | TE | TE |
| | Model 1 | Model 2 | Model 3 | Model 4 | Model 5 |
|---|---|---|---|---|---|
| lnFra | −0.015 * (−1.863) | 0.270 *** (6.613) | −0.065 *** (−8.802) | −0.021 ** (−2.534) | −0.021 *** (−2.617) |
| $(\text{lnFra})^2$ | 0.003 * (1.656) | −0.038 *** (−4.540) | | 0.004 ** (2.129) | 0.002 (1.440) |
| lnDP | | | | 0.021 *** (4.069) | |
| $(\text{lnDP})^2$ | | | | | |
| NF | | | | | 0.103 *** (2.934) |
| $\text{NF}^2$ | | | | | −0.193 *** (−5.846) |
| lnHA | −0.026 * (−1.652) | 0.070 (0.919) | 0.058 (1.329) | −0.027 * (−1.757) | −0.021 (−1.411) |
| HH | 0.011 *** (3.288) | 0.044 ** (2.566) | −0.01 (−1.003) | 0.010 *** (3.027) | 0.010 *** (2.959) |
| HE | 0.003 (0.794) | −0.043 ** (−2.553) | 0.015 (1.538) | 0.004 (1.066) | 0.005 (1.578) |
| lnFP | −0.009 (−1.08) | −0.116 *** (−2.677) | 0.156 *** (6.369) | −0.007 (−0.801) | 0.007 (0.850) |
| lnFL | 0.021 ** (2.076) | 0.181 *** (3.680) | 0.047 * (1.667) | 0.017 * (1.691) | 0.025 *** (2.637) |
| Di | −0.015 * (−1.768) | −0.001 (−0.027) | 0.016 (0.668) | −0.015 * (−1.775) | −0.013 (−1.581) |
| Co | 0.040 *** (4.19) | 0.055 (1.152) | −0.003 (−0.103) | 0.039 *** (4.089) | 0.038 *** (4.157) |
| VL | −0.011 (−0.369) | −0.135 (−0.891) | −0.076 (−0.889) | −0.008 (−0.277) | −0.018 (−0.630) |
| Low-altitude zone | 0.013 (1.449) | −0.109 ** (−2.417) | 0.113 *** (4.435) | 0.015 * (1.708) | 0.025 *** (2.832) |
| Medium-altitude zone | 0.027 *** (3.605) | −0.107 *** (−2.936) | 0.137 *** (6.647) | 0.029 *** (3.922) | 0.039 *** (5.432) |
| High-altitude zone | control | control | control | control | control |
| constant | 0.673 *** (9.968) | 0.056 (0.168) | 0.233 (1.231) | 0.672 *** (10.002) | 0.672 *** (10.427) |
| F | 5.62 *** | 10.39 *** | 24.87 *** | 6.52 *** | 16.25 *** |
| $R^2$ | 0.044 | 0.078 | 0.156 | 0.054 | 0.133 |

Note: t-values in parentheses, * indicates $p < 0.10$, ** indicates $p < 0.05$, and *** indicates $p < 0.01$.

**Table 8.** Results of the U test.

| Variable | The "U"-Shaped Relationship of Fragmentation on the Technical Efficiency | | Inverted "U"-Shaped Relationship between Fragmentation and Crop Diversification | | Inverted "U"-Shaped Relationship of Part-Time Farming on Technical Efficiency | |
| | Lower Bound | Upper Bound | Lower Bound | Upper Bound | Lower Bound | Upper Bound |
|---|---|---|---|---|---|---|
| Interval | 0.000 | 6.446 | 0.000 | 6.446 | 0.000 | 0.999 |
| Slope | −0.018 | 0.024 | 0.281 | −0.232 | 0.119 | −0.281 |
| t-value | −2.130 | 1.755 | 6.866 | −3.328 | 3.351 | −8.420 |
| $p > |t|$ | 0.017 | 0.040 | 0.000 | 0.001 | 0.001 | 0.000 |
| boundary | lnFra = 2.7 | Fra ≈ 15 | lnFra = 3.6 | Fra ≈ 36 | NP = 0.3 | |

## 4. Discussion

### 4.1. Analysis of the Impact of Land Fragmentation on Agricultural Technical Efficiency

The impact of land fragmentation on agricultural technical efficiency follows a weak "U" curve relationship. As land fragmentation increases, land plots become more dispersed in space, leading to lower efficiency in the use of fertilizers, pesticides, agricultural machinery, and labor input. This also results in higher agricultural production costs and a lack of rational allocation of agricultural production factors [29,46]. Consequently, technical efficiency can be significantly reduced. However, when the degree of fragmentation exceeds a critical value of Fra of 15, the scope economy effect outweighs the scale operation effect due to farmer planting diversification, leading to a slight increase in technical efficiency with increased fragmentation [36]. It is important to note that 66.02% of farmers in the study area operate with multiple land parcels below this critical value, indicating that the technical efficiency of most farmers is negatively affected by fragmentation. This highlights the need to reduce land fragmentation to enhance overall technical efficiency.

Meanwhile, significant variations exist in agricultural technology efficiency across different altitudes, with the high-altitude region having the lowest efficiency of 0.612. The reason for this is that high-altitude areas have low temperatures and harsh climatic conditions that are unsuitable for crop production [47], as well as inconvenient transportation,

which limits the transportation and sale of agricultural products, raises farmers' transaction costs [48], and reduces agricultural production efficiency. Additionally, the impact of land fragmentation on agricultural technical efficiency varies with different altitude levels. According to this study's findings, there is a significant negative effect of land fragmentation on technical efficiency at low altitudes (coefficient = −0.009, $p$ = 0.088), and the effect of land fragmentation on technical efficiency at medium altitude shows a "U" curve (primary coefficient of −0.024, quadratic coefficient of 0.005, which is significant at the 5% statistical level); there is no significant link between land fragmentation and technical efficiency in high-altitude zones, with a significance test of $p > 0.10$. The reason for this is that low-altitude areas are primarily planted with rice, wheat, corn, and other field food crops, so the production process necessitates a large number of mechanical operations, and the fragmentation makes it difficult for farmers to use mechanized services, thus reducing technology efficiency [32,33]. On the other hand, high-altitude areas are primarily planted with horticulture and other cash crops, with a low degree of mechanization. Although land fragmentation will have a negative effect on agricultural technical efficiency, crop diversification as a result of land fragmentation may improve technical efficiency, so that the net effect is positive [41]. Additionally, altitude has a negative effect on non-farm employment, with high-altitude areas showing a lower degree of labor market development compared to low-altitude areas [49]. Farmers have very few non-farm employment opportunities, and land fragmentation promotes intensive farming performance [19,20].

Therefore, the management of land fragmentation should focus on intermediate and lower elevations. In low-altitude regions, the primary reason for the huge number of plots and amount of land dispersion is the family contract responsibility system and other property rights system factors [50]. To address this issue, farmers should be encouraged to voluntarily participate in land swaps [11,51,52]. In particular, farmers who have primarily engaged in non-agricultural activities and have a high degree of part-time employment should be encouraged to transfer their farmland [53,54]. Additionally, the implementation of land improvement projects and other engineering measures should be employed to facilitate the consolidation of small fields into larger ones, thereby promoting large-scale and moderate management [12,55]. In the medium-altitude area, land fragmentation is mainly caused by the family contract responsibility system, topography, geomorphology, and other natural conditions, in addition to the land swaps, land transfers, and land remediation [46,56]. It can also be implemented by balancing the inward and outward movement of cropland, adjusting land use patterns that do not conform to the natural geographical conditions, achieving regional replacement and guiding the centralized distribution of land use [57].

### 4.2. Analysis of the Mediating Effects of Crop Diversification and Part-Time Farming

There are mediating effects of crop diversification and part-time farming on the impact of land fragmentation on agricultural technical efficiency. Specifically, there is an inverted "U" relationship between land fragmentation and crop diversification, with a critical value of Fra = 36. In the study area, 90.05% of households operate below this critical value, indicating that land fragmentation promotes crop diversification as farmers plant different crops based on the varying water and soil conditions of different land plots [21–23]. On the other hand, land fragmentation has a significant inhibiting effect on part-time farming, with a regression coefficient of −0.065. It increases farmers' agricultural labor time, reducing their leisure time and time available for non-agricultural work [25]. Furthermore, there is a significant positive linear relationship between crop diversification and agricultural technical efficiency, with a regression coefficient of 0.020. Crop diversification allows farmers to adaptively plant different crops based on soil and location conditions, thus improving technical efficiency [40,41]. Part-time farming, on the other hand, exhibits an inverted "U" relationship with agricultural technical efficiency, with a critical value of NP = 0.3. In the study area, 81.98% of households have a level of part-time farming above this critical value, indicating that as part-time farming increases, the labor force available for technical efficiency decreases. Additionally, with the migration of working-age laborers for

non-agricultural employment, the quality of labor force in agricultural production declines, leading to a decrease in agricultural technical efficiency [38,39].

Therefore, the government should guide farmers to choose appropriate farming practices such as intercropping and crop rotation based on the conditions of their land plots, which are ideal for diverse cultivation and increase agricultural technology efficiency [40]. Additionally, based on the principle of self-adaptation and common benefit, the government should encourage the active participation of relevant stakeholders [8] and strengthen employment guidance and skills training for farmers. For households with a high degree of non-agriculturalization, the government should provide employment services to reduce their reliance on cultivated land, which will facilitate the transfer of land to households with a greater dependence on farming [53,54]. For households that primarily rely on agricultural income, the government should provide them with enhanced training in cultivation skills and guide them towards becoming new types of agricultural operators [58].

*4.3. Deficiencies in Research*

This study aims to investigate the relationship between land fragmentation and agricultural technical efficiency, while also considering the mediating effects of crop diversification and part-time farming. The findings of this study provide valuable insights and recommendations. However, it is important to acknowledge the limitations of this research. The measure of land fragmentation used in this study was based solely on the number of plots operated by farmers. While this is a commonly used indicator, it fails to consider other important factors such as plot size or inter-plot distance. Therefore, future studies should consider constructing multidimensional indicators that incorporate these aspects to provide a more comprehensive understanding of land fragmentation. Another limitation of this study is the lack of an in-depth analysis on why the impact of land fragmentation on the agriculture technical efficiency varies in different regions. The Section 4 only briefly touched upon this aspect, citing other studies. Hence, further analysis and investigation of this topic should be undertaken in future studies from the perspective of property rights systems.

**5. Conclusions**

In this study, the SFA model was used to assess farmers' technical efficiency in the study area, and the Tobit and mediated effects models were used to examine the influence of land fragmentation on agricultural technology efficiency. The conclusions are as follows: (1) Cropland in Southern China is severely fragmented, and agricultural production has not reached the technological frontier, suggesting the existence of technological inefficiency. Furthermore, significant variations in the technical efficiency exist across different altitudes, with the lowest efficiency in high-altitude zones. (2) The impact of land fragmentation on agricultural technical efficiency encompasses both direct and indirect effects. Notably, the overall effect of land fragmentation on agricultural technical efficiency follows a "U" curve relationship, with a critical value of the curve at Fra = 15. In the study area, 66.02% of farmers operate with a number of land parcels below this critical value. As land fragmentation increases, technical efficiency initially improves but eventually declines. The technological efficiency is declining as the farmed land becomes more fragmented. (3) Crop diversification and part-time farming are mediating factors in the impact of land fragmentation on technical efficiency. Specifically, an "inverted U"-shaped relationship exists between land fragmentation and crop diversification, whereas a negative linear relationship is observed between land fragmentation and part-time farming. Conversely, crop diversification presents a positive linear relationship with technical efficiency, and part-time farming has an "inverted U"-shaped relationship with technical efficiency. (4) The impact of land fragmentation on agricultural technical efficiency varies by altitude, with a significant negative linear impact in low-altitude regions, a "U" curve relationship between land fragmentation and technical efficiency in medium-altitude regions, and no significant correlation between land fragmentation and technical efficiency in high-altitude regions.

As a result, the management of farmland fragmentation must be tailored to local conditions, with a focus on low- and medium-altitude regions. At the same time, the government should assist farmers in carrying out proper varied cultivation, improving the agricultural technology efficiency and strengthening employment guidance and skills training for farmers to ensure the orderly transfer of land.

**Author Contributions:** Conceptualization C.Z. and Y.Z.; methodology, C.Z.; software, C.Z.; validation, C.Z. and Y.Z.; formal analysis, C.Z.; investigation, C.Z. and M.L.; resources, X.L.; data curation, C.Z.; writing—original draft preparation, C.Z.; writing—review and editing, C.Z., Y.Z., M.L. and X.L.; visualization, C.Z. and Y.Z.; supervision, C.Z. and Y.Z.; project administration, Y.Z.; funding acquisition, Y.Z. All authors have read and agreed to the published version of the manuscript.

**Funding:** This research was funded by the Natural Science Foundation of Guizhou Province (No. [2021]YIBAN184) and the Project of the Science and Technology Innovation Base Construction of Guizhou Province (No. [2023]005).

**Data Availability Statement:** The data presented in this study are available on request from the corresponding author. The data are not publicly available due to privacy restrictions.

**Acknowledgments:** We sincerely thank the editors and reviewers who commented on this paper and gave their time and effort.

**Conflicts of Interest:** The authors declare no conflicts of interest.

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
