# Peer review of "How Does Land Fragmentation Affect Agricultural Technical Efficiency? Based on Mediation Effects Analysis"

_land, doi:10.3390/land13030284_

Round 1

Reviewer 1 Report

Comments and Suggestions for Authors

1)Line 23-25 "The results of the study 23 can provide lessons to promote the management of land fragmentation and improve the agricultural 24 technical efficiency" How , please explain 

2) Line 389-395: Does it is the author statement, if yes, please support it with your data, if no, a citation is needed.

3) Line 395-398: Can the give a justification for this from his data or literature 

4) Line 400-413: These line must be supported by the author own data or literature with a cited sources  

5) Section 4.2: Also  must be supported by the author own data or literature with a cited sources 

6) Kindly enrich your conclusion with the application and implication of your study

7) The author stated in the introduction section that as per available literature  land fragmentation and technical efficiency have either no, zero, or positive relation. And as per the author results "At lower altitudes, land fragmentation exhibits a notably negative effect on 395 technical efficiency. Conversely, at medium altitudes, the impact of land fragmentation  on technical efficiency follows a "U" curve relationship. However, no significant correlation is found between cropland fragmentation and agricultural production efficiency at 398 higher altitudes. In my opinion, land tenure system, secure and stable rights might have a considerable effect in this regard. I am suggesting that the author must also this. If it is not possible at this stage so the author may discuss this in the discussion section. Also he may include this in his limitation section. 

Author Response

Dear Reviewer:

    Thank you for your comments concerning our manuscript entitled “How does land fragmentation affect agricultural technical efficiency? The mediating role of crop diversification and part-time farming” (ID: land-2830022). These comments are very valuable for improving the manuscript and guiding the study. After careful study and consideration of these opinions, we have made detailed revisions to the paper according to your suggestions. Please refer to the attached document and the revised version that has been uploaded for specific modifications.

Thank you again for your valuable comments and best regards!

                                                                  Yours sincerely,

Authors

Reviewer 2 Report

Comments and Suggestions for Authors

This article analyzed the influence of land fragmentation on agricultural technical efficiency. The contribution is interesting. However, some points need to be better detailed for a complete understanding.

Title: a lot of words (18). Focus on the main aspects.

Keywords: Include different words in the title.

Abstract: Indicate year(s) of data and include more quantitative information/results.

Introduction: is comprehensive whit a good overview of problem in context. Include more details about the reason for defining this area of ​​study: 12 provinces in southern China, and improve the text of the study objective.

Materials and Methods: It is important to include more details, especially in the: 1. better describe the soils, climate, historical land use and land cover of the study provinces, 2. better justify the “research methodology” and “variable selection” (references in other studies), 3. indicate details of the period (year) of study and 4. indicate the geographic coordinates in figure 1.

Results: is correctly interpreted. Improve the resolution (visualization) of figures: 2, 3 and 4. Adjust the Table 2. (Z值 ???)

Discussions: is good, however, should be more detailed and with more references to other studies on the topic. They could better discuss and interpret the results from the perspective of previous studies and study hypotheses. For example: 1. It would be interesting to detail possible other influences (or not) of different period (year) of study, 2. Comment further on possible limitations of the variables analyzed, and 3. The study's applications for land use and cover analysis.

Conclusions: The text should focus more on the main results and highlight’s.

Author Response

(The authors gave the same response as above.)

Reviewer 3 Report

Comments and Suggestions for Authors

The authors have provided an intersting study of agricultural land fragmentation in China. There are a few things they need to add to clarify the research. First, in the Introduction, China is not able to produce enough food to feed all of its people. China imports a large amount of grain and soybeans, especially for animal feed. Second, the authors need to provide data on average and median farm size in the low, intermediate, and high altitude areas. Chinese farms are quite small on average, less than 3 hectares. The low altitude areas have much greater capacity for extensive crops (rice, corn, wheat), so fragmentation does not have good results there.

The literature review cites several studies in developing countries where access to land increases with fragmentation, which is viewed as a positive outcome. In developed countries, such as the US, where the dominant style of farming is extensive, fragmentation is seen as having negative results on farm technical productivity - See Daniels, McCarthy, and Lapping Society and Natural Resources, 2022.

The U-shaped and inverted U-shaped analyses are good. The inverted U-rela tionship test for land fragmentation and crop diversification underscores the value of mechanization on large farm parcels. The authors report: "cropland fragmentation had a statistically significant negative linear impact on agricultural technical efficiency in low-altitude regions.

Similarly, the inverted U-relationship test for part-time farming on technical efficiency points to the need for full-time farm operators as the parcel sizes increase.

Comments on the Quality of English Language

Minor editing needed.

Author Response

(The authors gave the same response as above.)

Round 2

Reviewer 1 Report

Comments and Suggestions for Authors

1) "17.395 plots of land" Would be better if the author specify the plot size .line 16 

2) Please enrich your introduction section with respect to studies related policies. Please refer to the articles"Hu, Q., Zhao, Y., Hu, X., Qi, J., Suo, L., Pan, Y.,... Chen, X. (2022). Effect of saline land reclamation by constructing the “Raised Field -Shallow Trench” pattern on agroecosystems in Yellow River Delta. Agricultural Water Management, 261, 107345. doi: https://doi.org/10.1016/j.agwat.2021.107345 and  Jiang, C., Wang, Y., Yang, Z., & Zhao, Y. (2023). Do adaptive policy adjustments deliver ecosystem-agriculture-economy co-benefits in land degradation neutrality efforts? Evidence from southeast coast of China. Environmental Monitoring and Assessment, 195(10), 1215. doi: 10.1007/s10661-023-11821-6

3) Please also your discussion section with respect to policy analysis please refer to 

Jiang, C., Wang, Y., Yang, Z., & Zhao, Y. (2023). Do adaptive policy adjustments deliver ecosystem-agriculture-economy co-benefits in land degradation neutrality efforts? Evidence from southeast coast of China. Environmental Monitoring and Assessment, 195(10), 1215. doi: 10.1007/s10661-023-11821-6

Author Response

Dear Reviewer:

    Thank you for your comments concerning our manuscript entitled “How does land fragmentation affect agricultural technical efficiency? The mediating role of crop diversification and part-time farming” (ID: land-2830022). These comments are very valuable for improving the manuscript and guiding the study. After careful study and consideration of these opinions, we have made detailed revisions to the paper according to your suggestions for the second time. Please refer to the attached document and the revised version that has been uploaded for specific modifications.

Thank you again for your valuable comments and best regards!

                                                                  Yours sincerely,

Authors

Reviewer 2 Report

Comments and Suggestions for Authors The authors adjust the text.  

Author Response

Dear Reviewer:

     we are pleased that our modifications have met your requirements, and we sincerely appreciate your valuable suggestions once again.

Thank you and best regards!

Yours sincerely,

Authors